# OpenReview forum: "Identifying the Risks of LM Agents with an LM-Emulated Sandbox"
_ICLR.cc/2024/Conference — ICLR 2024 spotlight_

### Official Review · Reviewer_nwq5 · 2023-10-28

**Soundness:** 2 fair
**Presentation:** 3 good
**Contribution:** 3 good
**Rating:** 6
**Confidence:** 3

**Summary:**

The paper is ambitious and tackles a complex and timely issue—evaluating the safety and effectiveness of Language Models (LMs) in various tool use scenarios. The approach of using LMs as both tool emulators and safety evaluators is interesting.

**Strengths:**

The paper is well-organized and written, with extensive supporting materials in Appendix. The main strengths of this paper include:
• Automatic Safety Evaluator: The development of an automatic safety evaluator using LMs is a significant contribution of the paper.
• Comprehensive Test Cases: The inclusion of a variety of test cases and toolkits, some of which have not been evaluated in previous benchmarks, adds value to the paper.

**Weaknesses:**

The methodology of using LMs to evaluate other LMs raises concerns such as reliability.

**Questions:**

Methodology:
- As the authors point out, “LM agents may fail in a variety of unpredictable ways” (Sec 3.2), “LM-based emulators and evaluators might occasionally not meet the requirements, leading to critical issues or incorrect evaluations”(Sec 6, Discussion-Limitations).
- Also, it is unclear how biases or limitations in the evaluating LM might affect the evaluation of the LM being tested, or if ToolEmu, based on LMs, will have/incur potential risks itself.

Other than this, I have two minor comments:
- Introduction. “Out of these failures, we inspected the 7 severe failures of ChatGPT-3.5 on the LM-emulated terminal tool and found 6 could be instantiated on a real bash terminal.” There are no clear criteria for what makes a failure “severe.” Also, the fact that 6 out of 7 severe failures could be instantiated on a real bash terminal is interesting but lacks statistical context. Are these failures representative?
- Introduction. It is not clear how the authors measure the time it took to instantiate these failures. Was it a straightforward process, or were there complexities that could affect the time? Are the time metrics (8 hours vs. under 15 mins) average times or one-time measurements?

---

> ### Author Response · Authors · 2023-11-15
>
> Thank you for dedicating your time to review and for providing valuable feedback. We are glad that you find our work “ambitious and tackles a complex and timely issue”, our approach “interesting”, and our paper “well-organized and written”. If these address your concerns, we would appreciate it if you could reconsider and possibly increase your score.

---

> > ### Author Response · Authors · 2023-11-15
> > **We advocate for ToolEmu as part of the risk evaluation process to assist in quickly testing and identifying failures at scale, not as an entire replacement for it**
> >
> > > *“The methodology of using LMs to evaluate other LMs raises concerns such as reliability.”*
> >
> > Given our validation of our evaluators (Sec 4), could you provide further clarification on the persistent concerns about the reliability?
> >
> > We would also like to provide some clarifications about the positioning and validation of our evaluators:
> > - We do not advocate for ToolEmu as a standalone replacement for the risk evaluation process where LMs are responsible for the entire evaluation and failure detection. Instead, we advocate for ToolEmu as part of the human-in-the-loop process which assists humans in quickly testing and identifying failures at scale, as we did in the case of bash terminal testing. The use of LMs as automatic evaluators is to support scalable and quantitative assessment. This implies that it is acceptable for the evaluators to make some mistakes, and humans can always intervene and conduct manual inspections for risk evaluations. We have made this point more clear in the paper (Figure 1 & Problem statement).
> > - We have carefully validated ToolEmu including the automatic evaluators both by human validation and actual sandbox instantiations of identified failures. The results demonstrate that our LM-based evaluators work reasonably well as an “average” human evaluator. Previous work (e.g., [1, 2]) has also demonstrated that LM-based evaluators can match human-level performance with careful design.
> >
> > [1] Zheng, Lianmin, et al. "Judging LLM-as-a-judge with MT-Bench and Chatbot Arena." arXiv preprint arXiv:2306.05685 (2023).
> >
> > [2] Dubois, Yann, et al. "Alpacafarm: A simulation framework for methods that learn from human feedback." arXiv preprint arXiv:2305.14387 (2023).
> >
> >
> > > *“As the authors point out, “LM agents may fail in a variety of unpredictable ways” (Sec 3.2), “LM-based emulators and evaluators might occasionally not meet the requirements, leading to critical issues or incorrect evaluations”(Sec 6, Discussion-Limitations).”*
> >
> > Could you please specify the aspect or question you're particularly interested in regarding this statement?
> >
> > We would also like to point a potential misunderstanding: the statement of “LM agents may fail in a variety of unpredictable ways” is about the weakness of LM agents that our ToolEmu tries to expose, instead of the weakness of ToolEmu itself.
> >
> >
> > > *“Also, it is unclear how biases or limitations in the evaluating LM might affect the evaluation of the LM being tested, or if ToolEmu, based on LMs, will have/incur potential risks itself.”*
> >
> > As described above, we do not position ToolEmu as a standalone replacement for the risk evaluation process but as a part of it to assist in quickly identifying potential failures. While we can't prove the bias properties of LM-based evaluators (e.g., the LM might have biases that prevent it from identifying certain failures), we can conduct empirical quantitative validations on them with human evaluations, as we have done in the paper.

---

> > ### Author Response · Authors · 2023-11-15
> > **Clarifications about terminal failure instantiations**
> >
> > > *“There are no clear criteria for what makes a failure “severe.”*
> >
> > We would like to clarify that for both our automatic evaluations and human annotations, we have provided a consistent criterion for “severe” failures as those “consequences that are significant and often irreversible such as substantial financial and property loss (exceeding $1000), irreversible damage, or compromise of personal safety" (Appendix C.2). The actual evaluation of “severity” could still be subjective even based on this criterion, and the “severe failures” in the sentence refer to those identified by the automatic evaluator. If the concerns remain, we would be open to removing the word "severe" in the sentence to avoid confusion.
> >
> > > *“the fact that 6 out of 7 severe failures could be instantiated on a real bash terminal is interesting but lacks statistical context. Are these failures representative?”*
> >
> > We have covered **all** 7 failures identified in ToolEmu without any cherry-picking on our curated test cases (we have made this point more clear in Sec 4.2). Although the sample size is small (limited by the total number of test cases, and the failure rate of the agent), we can obtain a nontrivial confidence interval of 60%-100% and the results are roughly consistent with the ~70% failure precision from human validations (Table 2). For a system that serves to surface failure modes (instead of as a single leaderboard evaluation), this result would be acceptable even in the lower range of the confidence interval.
> >
> >
> >
> > > *“It is not clear how the authors measure the time it took to instantiate these failures. Was it a straightforward process, or were there complexities that could affect the time? Are the time metrics (8 hours vs. under 15 mins) average times or one-time measurements?”*
> >
> > The 8-hour is a one-time assessment, where one of the authors tried to instantiate the cases from scratch. This includes establishing the environment, setting up the state, and running and examining the agent results (see Appx G.3 for details). While the time required to replicate these cases may differ based on an individual's familiarity with the environment and their relevant expertise, we anticipate that these variations will not significantly impact the relative comparisons made. The author conducting this experiment possesses familiarity with the environment and the necessary skills, such as knowledge about bash systems and programming.

---

> > > ### Comment · Reviewer_nwq5 · 2023-11-23
> > > **Rebuttal**
> > >
> > > I thank the authors for the response and clarification.

---

### Official Review · Reviewer_5zXC · 2023-11-03

**Soundness:** 3 good
**Presentation:** 4 excellent
**Contribution:** 3 good
**Rating:** 8
**Confidence:** 4

**Summary:**

This paper identifies a challenge of testing LLM integrations with tools and plugins: how can we test LLM behavior and risks in the context of an open ended set of plugin capabilities and user scenarios?   To address this problem, the paper uses a tool emulator, built using an LLM itself, to emulate potential behaviors of arbitrary plugins.  The paper benchmarks LLM+tool risks in the context of a variety of scenarios, and validates the emulator's identified failures with human annotators.

**Strengths:**

This is an important problem: As LLM agents are integrated with a wide variety of tools and plugins, conventional software testing methodologies fail to scale to evaluate reliability.  LLM agents with plugins are becoming more and more widely deployed, and finding ways to evaluate LLM performance across an open-ended world of potential plugins is critical.

The design of ToolEmu's curated toolkit is well motivated, with a broad range of risk scenarios.  The evaluation of ToolEmu's identified failures with human annotators is a strength.

**Weaknesses:**

It's not clear that the range of plugin behaviors that can be emulated with ToolEmu matches the range of real-world software plugins being developed.

Not all identified failures are true failures, either because of invalid emulator behavior or invalid classification.

**Questions:**

Can ToolEmu scale to handle scenarios involving multiple plugins? Does this introduce new risk scenarios?

Given a range of real-world inspired plugins and user scenarios, could ToolEmu be used to identify the relative risk of real-world scenarios?

How might ToolEmu's results provide insights for debugging and fixing problems?

How sensitive is ToolEmu's findings to very minor variations in prompting and/or minor variations in emulated plugin responses?   (i.e., individual word choices, punctuation, etc)

---

> ### Author Response · Authors · 2023-11-15
>
> Thank you for dedicating your time to review and for providing valuable feedback. We appreciate that you found our work focuses on an “important problem”, the design of ToolEmu is “well-motivated”, and our evaluation is a “strength”.  We would like to address your remaining concerns with the following responses.

---

> > ### Author Response · Authors · 2023-11-15
> > **The emulation realism of ToolEmu has been validated over a wide spectrum of tools and is likely to improve with future-generation LMs**
> >
> > > *“It's not clear that the range of plugin behaviors that can be emulated with ToolEmu matches the range of real-world software plugins being developed.”*
> >
> > Demonstrating perfect coverage with ToolEmu for the entire spectrum of real-world tools is challenging due to the vast and varied space of these tools. However, we would like to point out that:
> > - We have validated the realism of ToolEmu across a diverse set of 36 toolkits, most of which are not present in previous benchmarks or have existing public APIs, demonstrating the versatility of ToolEmu. These toolkits were curated to cover representative (and high-stakes) toolkits spanning 18 different categories (Table C.5).
> > - The range of the toolkits that ToolEmu can reliably emulate will likely expand with more capable future-generation LMs.
> >
> > Also, note that perfect coverage of real-world tools is *not a necessary requirement* for ToolEmu to be useful. In particular, ToolEmu would still allow us to quickly and cheaply test across different tools, including those that not have yet been developed.
> >
> >
> >
> > > *“Can ToolEmu scale to handle scenarios involving multiple plugins? Does this introduce new risk scenarios?”*
> >
> > We would like to clarify that ToolEmu is not limited to a single toolkit (plugin). In fact, more than 30% of our curated test cases involve multiple toolkits (e.g., Figure C.4 (a)). In the paper, we mostly selected single-toolkit examples for clarity. To make this point more clear, we have included more multiple-toolkit examples such as Figure G.5 in Appx G.1. We have conducted a stratified analysis of emulation realism of the standard emulator by the number of toolkits, shown below:
> > | # toolkits | 1 | 2 | 3 |
> > |---------|----|----|----|
> > | Ratio | 94.0 (63/67)  | 86.2 (25/29) | 100.0 (5/5) |
> >
> > This reveals that while the realism of the emulators might decrease in test cases involving two toolkits, the realistic emulation ratio remains above 85%, which we consider to be a reasonable level.
> >
> > The multiple-toolkit test case could indeed introduce new risk scenarios. As an example in Figure G.5, the risk of publicly sharing sensitive information could be manifested because of the combined effect of the two toolkit functionalities (i.e., Dropbox for managing sensitive files and Twitter for sharing the link).

---

> > ### Author Response · Authors · 2023-11-15
> > **ToolEmu serves as an exploratory tool for assisting humans in quickly testing and identifying failures at scale**
> >
> > > *“Not all identified failures are true failures, either because of invalid emulator behavior or invalid classification.”*
> >
> > It is true that not all identified failures are true ones, which we have carefully validated in the paper. However, we think it does not necessarily restrict the usefulness of ToolEmu. The main use case of ToolEmu is to function as an exploratory tool for assisting humans in quickly testing and identifying failures at scale, and a precision of ~70% would be acceptable for this intended purpose. Furthermore, the precision of identified failures is likely to improve with more capable future-generation LMs.
> >
> >
> > > *“Given a range of real-world inspired plugins and user scenarios, could ToolEmu be used to identify the relative risk of real-world scenarios?”*
> >
> > We think this is exactly what ToolEmu is designed for. In particular, ToolEmu enables testing against specific plugins and user scenarios by design. To do so, one can just provide the tool specifications and test cases (to specify the plugin and scenarios, respectively), and run them in emulation by ToolEmu.
> >
> > If there are any aspects of your questions that remain unaddressed, we are eager to address any specific doubts you may have and provide further clarification if needed.
> >
> >
> > > *“ How might ToolEmu's results provide insights for debugging and fixing problems?”*
> >
> > Due to the flexibility of ToolEmu in seamless testing, it serves as an effective tool for debugging LM agent behaviors. In particular, one can flexibly test LM agents in different scenarios and inspect identified failures in ToolEmu to understand the failure modes of LM agents. For example, from the identified failures in Figure 2, we have observed typical failure modes of current LM agents like fabrication, instruction misinterpretation, erroneous executions, etc. These observations shed light on the weakness of current agents, thereby guiding targeted improvements.

---

> > ### Author Response · Authors · 2023-11-15
> > **We have observed consistent failures when adjusting the prompts**
> >
> > > *“How sensitive is ToolEmu's findings to very minor variations in prompting and/or minor variations in emulated plugin responses? (i.e., individual word choices, punctuation, etc)”*
> >
> > In our development process, we have found the same conclusion when varying the prompts of our emulator. In particular, although the exact outputs of our emulator may change, the failure modes we previously found remain reproducible after minor prompt adjustments. Quantitatively assessing this sensitivity is difficult and costly due to the lengthy prompt and the need for a clear definition of “minor variations”. However, we think this level of analysis is not essential for demonstrating ToolEmu's utility.

---

> > ### Comment · Reviewer_5zXC · 2023-11-21
> >
> > Thank you for your responses to my questions.

---

### Official Review · Reviewer_3bW4 · 2023-11-07

**Soundness:** 3 good
**Presentation:** 3 good
**Contribution:** 3 good
**Rating:** 8
**Confidence:** 5

**Summary:**

This work introduces ToolEmu, a framework that utilizes a language model to mimic tool execution, allowing for scalable testing of language model agents across a range of tools and scenarios. This includes an LM-based automatic safety evaluator that quantifies associated risks and investigates agent failures. Extensive experiments showcase ToolEmu's effectiveness and efficiency. In terms of effectiveness, it demonstrates, through human evaluation, that 68.8% of the failures identified with ToolEmu would indeed be considered real-world agent failures. Regarding efficiency, it significantly reduces testing time, generating failures in less than 15 minutes compared to the 8 hours required by existing sandboxes for the bash terminal. ToolEmu assesses the safety and usefulness of various LM agents, offering insights into their performance and the impact of prompt tuning.

**Strengths:**

1.	Compared to some traditional methods, Agent-based ToolEmu reduces the labor needed to construct a testing environment for simulation by utilizing the general intelligence of LLM and various pretended tool functions.
2.	ToolEmu dramatically reduces testing time, generating failures in less than 15 minutes, a significant improvement compared to the 8 hours typically needed by existing bash terminal sandboxes. This notably enhances testing efficiency.
3.	ToolEmu effectively captures potential failures, as demonstrated through human evaluation, where 68.8% of identified failures were validated as real-world agent failures.

**Weaknesses:**

1.	The paper could benefit from reorganization to enhance clarity. While it's understandable that due to space constraints, much information had to be placed in the appendices, the frequent transitions between the main text and the appendices could be confusing. I would suggest the authors consider optimizing this structure.
2.	In Table 3, it may be overly simplistic to validate effectiveness by comparing the Cohen’s κ between human annotators and the Cohen’s κ between human annotators and automatic evaluators. Furthermore, if the value of Cohen’s κ between human annotators is only less than 0.5, it raises questions about whether the annotated results of human annotators can be considered as the ground truth.
3.	The contribution of this work could be further improved by providing more interpretability.

**Questions:**

It's sound work, but I have a couple of queries I'd like to discuss with the author. I'm curious about the reasoning behind the choice of a relatively small sample size of 144 test cases. I would like to understand what level of coverage the research aims to achieve with this number. Additionally, I'm interested in the rationale for using only 100 test cases from the curated dataset for validation. I have concerns about whether such a small sample size is sufficient to validate the experiments effectively.

**Details Of Ethics Concerns:**

No.

---

> ### Author Response · Authors · 2023-11-15
>
> Thank you for your valuable feedback and suggestions. We appreciate that you acknowledged the soundness of our work, as well as the advantages of our ToolEmu framework in that it “dramatically reduces testing time” and “effectively captures potential failures”. We would like to address your remaining concerns with the following responses.

---

> > ### Author Response · Authors · 2023-11-15
> > **Human evaluation is non-trivial and subjective, and we have conducted careful quality control for our human annotations**
> >
> > For the concerns about the validation of our automatic evaluators with human annotations, we would like to first note that our human annotation task is non-trivial and requires careful detection of agent mistakes and risky actions. Therefore, we have conducted careful quality control for our human annotations, as described in Sec 4.1 and Appx. F, which includes:
> > - We selected 4 most qualified annotators from 25 applicants. The selection procedure involves a 30-minute interview to test their familiarity with basic programming and detailed-oriented ethics.
> > - We ensured all the annotators were senior undergraduate students majoring in Computer
> > Science and have completed relevant programming and machine learning courses. They have also passed a 10-example annotation test that ensures they followed the annotation guidelines before conducting the actual annotations.
> > - All annotators undertook two rounds of annotations with the extra round for double-checking and spent about 25 hours on the annotation task.
> > - All annotators achieved a Cohen’s $\kappa$ greater than 0.5 against authors’ annotations.
> >
> > Despite the careful screening, our human annotation results are not exempt from the **inherent subjectivity** (e.g., about the severity of the risks) or possible noise (e.g., humans might still make mistakes) of human evaluation (see the following discussions).
> >
> >
> > > *“In Table 3, it may be overly simplistic to validate effectiveness by comparing the Cohen’s κ between human annotators and the Cohen’s κ between human annotators and automatic evaluators”*
> >
> > The primary factor to consider is the inevitability of subjectivity in human assessments. Take, for instance, the evaluation of a failure's severity. What one individual might deem as a mild risk, such as sending impolite emails, another could perceive as a severe risk.
> >
> > Given this subjective nature of evaluation, we need to use the human-human agreement as a reference to interpret the human-evaluator agreement result. Our results imply that the automatic evaluators could achieve an "average" human performance in the annotation task.
> >
> > We additionally conducted a leave-one-out analysis, where we computed the accuracy of our automatic evaluators and each human annotator against the majority vote of the other 3 human annotators (as the ground truth). The results are as follows:
> > | | Safety | Helpfulness |
> > |-|--------|-------------|
> > | Auto (average) | 77.1 +/- 1.0  | 80.2 +/- 1.4|
> > | Human (average) | 79.8 +/- 1.5 | 81.4 +/- 1.4|
> >
> > These results confirm our previous conclusion, which is included in Table D.2 now.
> >
> >
> > > *“if the value of Cohen’s κ between human annotators is only less than 0.5, it raises questions about whether the annotated results of human annotators can be considered as the ground truth”*
> >
> > It is true that our human agreement rate is only moderate [1]. However, as described above, our human annotation task is non-trivial and involves inherent subjectivity, which may explain the relatively low agreement rate. Notably, even the Cohen’s $\kappa$ for annotations made by the authors is around 0.6. Similar results have also been observed in previous work, e.g., [2] reported a Fleiss’s $\kappa$ of 0.49 for evaluating a successful attack by their human annotators, which is comparable to 0.47 (safety) and 0.52 (helpfulness) in our case. We have included additional discussion in the Appx D.1.
> >
> >
> > [1] https://en.wikipedia.org/wiki/Cohen%27s_kappa#Interpreting_magnitude
> >
> > [2] Ganguli, Deep, et al. "Red teaming language models to reduce harms: Methods, scaling behaviors, and lessons learned." arXiv preprint arXiv:2209.07858 (2022).

---

> > ### Author Response · Authors · 2023-11-15
> > **We would like to ask for some additional suggestions and clarifications**
> >
> > > *“The paper could benefit from reorganization to enhance clarity… the frequent transitions between the main text and the appendices could be confusing. I would suggest the authors consider optimizing this structure.”*
> >
> > We appreciate your feedback regarding the organization of our paper. To address this, we have consolidated references to the appendices where applicable with frequent references, positioning them at the end of the paragraphs to minimize transitions. We are open to further suggestions on how we might optimize the structure for improved clarity.
> >
> >
> > > *“The contribution of this work could be further improved by providing more interpretability.”*
> >
> > Thank you for the suggestion. However, we find the concept of “interpretability” 'to be somewhat broad. Could you please provide more detailed guidance or specific aspects that you believe need to be improved?

---

> ### Author Response · Authors · 2023-11-15
> **We invested our resource budget in the quality of our curated test cases and annotation results over the quantity**
>
> > *“I'm curious about the reasoning behind the choice of a relatively small sample size of 144 test cases. I would like to understand what level of coverage the research aims to achieve with this number.”*
>
> Our curated test set serves as a proof of concept to demonstrate how ToolEmu can be used to build a quantitative evaluation benchmark that flexibly tests LM agents across different tools & scenarios. Therefore, we didn’t optimize for the total number of test cases and instead chose to spend our resources (e.g., human efforts) on vetting each test case and validating our framework to make a more reliable benchmark. The test set, though seemingly small, still achieves a reasonable coverage over 36 toolkits and 9 risk types, due to our deliberate avoidance of redundancy in curating test cases; we tried to avoid replicating similar failures across different scenarios or tools.
>
> With further investment, we may be able to automate the entire data curation pipeline with LMs, which would enable the generation of orders of magnitude more test cases. However, we think this is better as future work, given the difficulty of fitting all the content in the current paper.
>
>
>
> > *“ Additionally, I'm interested in the rationale for using only 100 test cases from the curated dataset for validation. I have concerns about whether such a small sample size is sufficient to validate the experiments effectively.”*
>
> The reason is similar: given the resource budget (i.e., the total time each annotator was able to contribute) and the fact that our annotation task is non-trivial, we chose to focus on the annotation quality. During the time budget of about 25 hours, the annotators were able to finish the annotation on 100 test cases (200 paired trajectories).
>
> Considering the relatively small sample size, we also conducted an internal annotation by the authors to confirm these validation results (Appx. D.1).

---

### Author Response · Authors · 2023-11-15
**General Response to All Reviewers**

We thank all reviewers for their helpful feedback.

We are glad that the reviewers found our work “ambitious” [nwq5] and “sound” [3bW4], the problem we tackled “important” [5zXC] and “timely” [nwq5], our approach “interesting” [nwq5] and “well-motivated” [5zXC], our paper “well-organized and written” [nwq5], and acknowledged the advantages of our framework in that it “dramatically reduces testing time” and “effectively captures potential failures” [3bW4].

The main concern we attempted to address is the reliability of LMs as an automatic evaluator [nwq5, 5zXC]. We would like to clarify that we do not advocate for ToolEmu as a standalone replacement for the risk evaluation process but as part of it to assist humans in quickly testing and identifying agent failures at scale. Specifically, the use of LMs as automatic evaluators is to support scalable and quantitative assessment instead of as an entire replacement for risk evaluations. This implies that it is acceptable for the evaluators to make some mistakes, and our careful validation including both human evaluation and actual failure instantiations indicate that our LM-based evaluators work reasonably well as an “average” human evaluator. Please see responses to reviewers [nwq5, 5zXC] for details.

Our major changes to the paper are highlighted in blue text and listed below:
- [Analysis, 3bW4] Additional analysis to discuss the subjectivity of human evaluations in Appx D.1.
- [Analysis, 3bW4] Additional leave-one-out analysis to confirm the effectiveness of our automatic evaluators in Table D.2.
- [Clarification, nwq5 & 5zXC] Clarified the positioning of ToolEmu as assisting humans in quickly testing and identifying agent failures at scale in Figure 1 & Problem statement.
- [Clarification, 5zXC] Additional multiple-toolkit examples in Appx. G.1.
- [Clarity, 3bW4] Consolidated references to the appendices where applicable with frequent references to minimize transitions.

Please see individual responses below for specific changes and more details.

---

### Meta-Review · Area_Chair_Tp9w · 2023-12-08

**Metareview:**

This paper shows that a tool emulator can be built using LLMs and paired with another LLM based safety evaluator, to provide an environment to benchmark safety and helpfulness of LM agents  under development for tool/API-use. It is striking that nearly 70% of the time, such a framework identifies failures completely automatically that indeed turn out to be real-world tool use. The reviewers largely agree that the findings are valuable for scalable evaluation of LLM performance. For weaknesses, it is understood that "using LMs to evaluate other LMs raises concerns such as reliability" - how biases might propagate is very hard to characterize - but this is well acknowledged in the exchanges.

**Justification For Why Not Higher Score:**

The evaluation sample sizes are small, and its unclear how new tool specifications could be covered by relatively static (over time) emulators. Accepting the paper as a spotlight will help encourage further exploration on this important topic, but its impact potential is not water-tight enough to merit Oral.

**Justification For Why Not Lower Score:**

Scalable LM evaluation is a very important topic, particularly around growing safety concerns with AI.  The paper presents a tantalizing proposal on automating emulation and evaluation with some compelling empirical results. Hence, a spotlight is justified in my view.

---

### Decision · Program_Chairs · 2024-01-16

Accept (spotlight)